# Coaggregation of Asthma and Type 1 Diabetes in Children: A Narrative Review

**DOI:** 10.3390/ijms22115757

**Published:** 2021-05-28

**Authors:** Laura Sgrazzutti, Francesco Sansone, Marina Attanasi, Sabrina Di Pillo, Francesco Chiarelli

**Affiliations:** Department of Pediatrics, University of Chieti-Pescara, 66100 Chieti, Italy; laura.sgrazzutti@studenti.unich.it (L.S.); francesco.sansone001@studenti.unich.it (F.S.); sabrinadipillo@gmail.com (S.D.P.); chiarelli@unich.it (F.C.)

**Keywords:** asthma, type 1 diabetes, children, coaggregation, cytokines, Th1/Th2 paradigm, hygiene hypothesis, genetic, regulatory T lymphocytes

## Abstract

Asthma and type 1 diabetes mellitus (T1DM) are two of the most frequent chronic diseases in children, representing a model of the atopic and autoimmune diseases respectively. These two groups of disorders are mediated by different immunological pathways, T helper (Th)1 for diabetes and Th2 for asthma. For many years, these two groups were thought to be mutually exclusive according to the Th1/Th2 paradigm. In children, the incidence of both diseases is steadily increasing worldwide. In this narrative review, we report the evidence of the potential link between asthma and T1DM in childhood. We discuss which molecular mechanisms could be involved in the link between asthma and T1DM, such as genetic predisposition, cytokine patterns, and environmental influences. Cytokine profile of children with asthma and T1DM shows an activation of both Th1 and Th2 pathways, suggesting a complex genetic-epigenetic interaction. In conclusion, in children, the potential link between asthma and T1DM needs further investigation to improve the diagnostic and therapeutic approach to these patients. The aim of this review is to invite the pediatricians to consider the potential copresence of these two disorders in clinical practice.

## 1. Introduction

Atopic diseases represent one of the major sources of morbidity in both adults and children worldwide. Atopy is the development of adverse hypersensitivity immune reactions against environmental antigens, usually mediated by immunoglobulin (Ig) E action, leading to several diseases such as atopic dermatitis, allergic rhinitis and conjunctivitis, asthma, and food allergy [1]. Asthma is a common chronic inflammatory disease characterized by symptoms such as wheezing, cough, shortness of breath, and variable expiratory airflow limitation that has a significant impact on quality of life of both adults and children [2]

The prevalence of child-onset asthma steadily increased in the second half of the 20th century and was higher in children living in more affluent countries. Recently, an important increase was also registered in developing countries [3,4]. Phase III of the International Study of Asthma and Allergy in Childhood (ISAAC) is the most comprehensive evaluation of the worldwide prevalence of asthma. It pointed out a global prevalence of current wheezing of 11.5% in the 6–7-year-old group and 14.1% in the 13–14-year-old group. The highest prevalence of asthma was reported in English-language countries and Latin America, while the disease seemed to be less often recognized but more severe in Africa, the Indian subcontinent, and the Eastern Mediterranean [3].

According to the National Health Interview Survey data, childhood asthma prevalence increased from 2001 to 2009. This increase was first followed by a plateau and then by an initial decline in 2013. Asthma prevalence trends have changed in relation to different sociodemographic conditions [5].

Concurrent to the increasing of the incidence of atopic diseases, there was also a substantial rise of autoimmune disorders [6]. Type 1 diabetes mellitus (T1DM) is a multifactorial chronic disease caused by a cell-mediated immune destruction of pancreatic β cells resulting from a complex interaction between genes and environmental factors [7]. T1DM incidence has steadily increased worldwide over the last few decades [8]. In a recent metanalysis of 193 articles, the worldwide incidence of T1DM was 15 per 100,000 individuals and the prevalence was 9.5 per 10,000 individuals [8]. There is a wide range of variation in the incidence of childhood T1DM among different geographical areas based on ethnic and racial distribution. A very high incidence (> or =20/100,000 per year) was documented in Sardinia, Finland, Sweden, Norway, Portugal, U.K., Canada, and New Zealand. On the other hand, the lowest incidence (<5/100,000 per year) was found in China, Japan, and Rwanda [9]. In general, the incidence of T1DM increased with the age, reaching a peak in 10–14-year-old children [9].

T1DM shares this increasing trend in children with asthma and atopic diseases, suggesting potentially common environmental factors and genetic susceptibility [10].

Starting from these epidemiological data, research groups have studied the relationship between these two disorders worldwide. The aim of this narrative review is to collect and critically discuss the scientific evidence which highlights the “possible link” between asthma and T1DM in childhood explaining the potential underlying molecular mechanisms.

## 2. Divergent Theories Underlying Asthma and Type 1 Diabetes Topic

### 2.1. Th1/Th2 Paradigm: Are Asthma and T1DM Mutually Exclusive?

Asthma and T1DM are chronic inflammatory diseases, although they classically involve opposite arms of the adaptive immune system: [11].

Th1 cells defend against infections and tumors but are also involved in the development of autoimmune diseases (T1DM), producing cytokines such as interferon (INF)-γ, Tumor Necrosis Factor (TNF)-α, and interleukin (IL)-2 [11].Th2 cells protect against parasites but also promote the development of IgE-mediated atopic diseases (asthma), producing cytokines such as IL-4, IL-5 IL-6, IL-9, IL-10, and IL-13 [11].

Several studies have sustained the so-called “Th1/Th2 paradigm” [12,13,14,15]. The theory, described three decades ago for the first time, explains that an expansion of the Th1 clones in individuals with T1DM would cause a reduction of the Th2 compartment, preventing the development of atopic diseases and vice versa [16]. According to this latter concept, autoimmune (Th1-related) and allergic diseases (Th2-related) would be mutually exclusive.

Consistent with the Th1/Th2 paradigm, several observational studies have suggested that asthma and allergic respiratory symptoms are reduced in patients with T1DM [12,13,14,15]. In addition, this protection seems to extend to nondiabetic siblings [15], suggesting an effect mediated by a shared genetic background or by the exposure to the same environmental factors during pregnancy or in early life. Tosca et al. [17] evaluated the lung function of 20 T1DM children with allergic rhinitis compared to 59 controls affected by allergic rhinitis alone. The authors found that only children with allergic rhinitis had a significant increase of forced expiratory flow at 25% and 75% of forced vital capacity after bronchodilation compared to the T1DM group, suggesting a protective role of T1DM associated with allergic rhinitis on asthma development. In a meta-analysis of 25 studies from Europe and North America, Cardwell et al. [18] described a reduction of frequency of asthma in children with T1DM (OR 0.82 CI 95% 0.68–0.99 *p* = 0.04). It is worth noting that this protective effect was only significant after the elimination of those studies with an inadequate design. Data also suggested an inverse association with eczema (OR 0.82 CI 95% 0.62–1.10 *p* = 0.18), although not statistically significant. A recent metanalysis of six studies suggested a potential protective role of diabetes against atopic dermatitis development (odds ratio, 0.69, 95% CI, 0.67–0.72) [19]

To date, the epidemiologic data are not certain enough to confirm the Th1/Th2 paradigm as a potential theory explaining the relationship between asthma and T1DM.

### 2.2. The “Hygiene Hypothesis”: Could Asthma and T1DM Have a Common Environmental Background?

Environmental factors are also important in the development of atopic diseases and T1DM [6]. The “Hygiene Hypothesis” is a theory suggesting that reduced or delayed exposure to infections at early ages may increase the risk of the onset of autoimmune disorders (atopic diseases or T1DM) [20]. Several studies have shown an increased incidence of asthma and other atopic diseases in the general population, particularly in patients with T1DM [21,22,23,24,25].

Klamt et al. [22], in a prospective population-based case-control study, demonstrated that children with T1DM reported the presence of IgE-mediated allergies more frequently than controls. Another cross-sectional study in Brazil [23] found that patients with T1DM had a higher prevalence of allergic diseases and atopic sensitization than expected. Hsiao YT et al. [24] reported a higher incidence of asthma in young patients with T1DM than in the general population in Taiwan, showing a negative effect of poor glycemic control on the risk of asthma development. In the United States, Black et al. [25] showed a higher prevalence of asthma (10.8%) in adolescents with T1DM compared to the general population (8.7%). However, these data should be interpreted with caution due a high percentage of obese and overweight participants in both T1DM and T1DM-asthma groups [26]. In this latter study, being overweight, a noted risk factor for asthma, could have acted as a confounding factor, making conclusions unreliable [27].

Fsadni et al. [21] compared the reported country incidence of T1DM (from the Diabetes Mondial Project Group [28]) to the prevalence of atopic diseases (from The ISAAC phase 1 study [29,30,31]) and found that T1DM had a positive correlation with both wheezing and atopic eczema. Stene et al. [32] showed a strong positive association between the occurrence of T1DM and asthma, analyzing epidemiological data extracted from three international studies (ISAAC 1998 [31], EURODIAB ACE Study Group 2000 [33], and Diabetes Mondial Project Group 1993 [28]). In particular, the authors used the published data on the prevalence of asthma symptoms (>four episodes of wheezing in the last 12 months in 13–14-year-old age group) and on the incidence of T1DM (among children aged 0–15 years old), obtained from the average of all centers in each country, and calculated the Spearman correlation coefficient. The authors suggested that common environmental factors might influence the susceptibility to both disorders [32].

The proposed molecular mechanisms underlying the “Hygiene Hypothesis” focus on the role of regulatory T lymphocytes (Treg) in the immune system response. The action and maturation of these cells can be modulated by several factors such as infectious agents, gut microbiome changes, and parasitic infections [34,35].

In the following subsections, the contribution of the aforementioned factors are discussed.

#### 2.2.1. Infections

Protective effects of bacteria (*Listeria monocytogenes*, *Propionibacterium acnes*, *Chlamydia*, *Lactococcus species*) or bacterial products on asthma development have been well documented [20,36]. Viruses also seemed to have a protective role against asthma in murine models [37]. Bacterial infections protect against asthma development by stimulating innate immune receptors, such as Toll-like receptors (TLR), and inducing Th1 responses [20]. Viral infections exert their action against asthma development by induction of a natural killer (NK) cell subset or monocytes with a regulatory phenotype [20,37].

On the other hand, recurrent wheezing episodes in early life are known to be a major risk factor for asthma development, and viral infections have been linked to asthma development in 62–98% of cases [38]. Respiratory Syncytial Virus- or Rhinovirus-induced wheezing in preschool children with family history of asthma were associated with an increased risk of asthma at 6 years of age [20].

Children living with several older siblings or attending daycare centers before 1 year of age were at lower risk of developing allergic disorders than children attending from 2 years onward and children from small families [39,40].

Similar findings have also been reported for T1DM [9]. Cardwell et al. [41] demonstrated a reduced risk of T1DM in children living with siblings, sharing a bedroom, and moving to a new house more often. Furthermore, living in a farm environment seems to be a protective factor against T1DM due to early exposure to a larger number of microorganisms and different farm animals [42]. These findings support the “Hygiene Hypothesis”, suggesting that the exposure to infections in early life may protect against the onset of T1DM.

#### 2.2.2. Gut Microbiome

Early-onset autoimmune diseases are common in Finland and Estonia but less prevalent in Russia [43]. Furthermore, sensitization to allergens and allergic symptoms are much more common in Finnish than in Russian schoolchildren [44].

Interestingly, evaluating the intestinal colonization in Finnish, Estonian, and Russian populations, the presence of Bacteroides species in the intestinal microflora was more frequent in Finnish and Estonian infants than in Russian infants. Therefore, they were exposed primarily to Lipopolysaccharide (LPS) of Bacteroides than to LPS of *E. coli*. The structure of the Bacteroides LPS, differently from *E. coli* LPS, inhibits the innate immune activation and endotoxin tolerance [43]. This could suggest the existence of a link between the composition of the intestinal microflora and the development of both diseases.

Gut microbiota composition in the first month of age is crucial for the development of a healthy immune system, and any alterations during this temporal window could affect its development irreversibly [45].

Interestingly, gut microbiota analysis in children at high risk of developing T1DM or asthma showed common peculiar taxonomic changes. These children had a lower biodiversity [46,47], a higher Bacteroides/Firmicutes ratio, a relative abundance of Clostridia, and a relative deficit of Lactobacillus and Bifidobacterium [46,48]. The mechanism underlying this phenomenon is largely unknown, but one possibility is represented by the production of biomolecules capable of interaction even at great distance from the bowel. Among them, small-chain fatty acids (SCFAs) seem to be particularly promising. SCFAs, namely acetate, butyrate, and propionate, are produced by several different bacteria in the gastrointestinal tract through the fermentation of fibers [49]. They contribute to the regulation of both the innate and adaptive immune system by the G-protein coupled receptor 43 (GPR43). Dietary supplementation with butyrate and acetate in non-obese diabetic (NOD) mice had a protective effect by reducing the incidence of autoimmune diabetes and delaying its onset [50]. Similarly, mice with a defect of GPR43 or SCFAs production exhibited a stronger inflammatory response after exposure to common aeroallergens, showing higher production of Th2 proinflammatory cytokines [51].

Acetate decreases autoreactive T cells while butyrate promotes Treg differentiation and function [52]. In addition, they downregulate the major histocompatibility complex (MHC) class I and costimulatory proteins expression on B cells and promote differentiation of B cells into plasma cells and memory cells capable of producing specific IgG and IgA [53]. Furthermore, acetate and propionate improve insulin sensitivity, whereas butyrate maintains the integrity of gut epithelium [54].

#### 2.2.3. Parasitic Infections

The improved living conditions in developed countries have also caused a decline of parasitic infections. This seems to be correlated to an increase of the incidence of immune-mediated disorders and atopic diseases [55,56]. According to a recent review, helminths could prevent the development of autoimmune and atopic diseases [57]. Helminths seem to regulate both the innate and adaptive immune systems, promoting a typical Th2 response and modulating Th1/Th17 differentiation, causing an increase of Th2-related cytokines and a reduced secretion of Th1/Th17-related cytokines [58].

In addition, parasite-derived proteins can also modulate the bacterial presence of the gut microbiota, leading to indirect regulation of the immune system [59]. Immunomodulation induced by helminths may prevent diabetes mellitus and improve insulin sensitivity [60,61]. Products from helminths, such as *Fasciola hepatica* helminth defense molecule (FhHDM) [62] and Omega-1 of *S. mansoni* [63], play an important role against atopic diseases. However, the link of atopic and autoimmune disorders with helminth infection is still controversial. The aforementioned environmental factors may explain the common epidemiologic trend of asthma and T1DM, although they have immunological differences, as expressed by the Th1/Th2 paradigm.

### 2.3. Genetic Protective or Risk Factors for Asthma and Type 1 Diabetes

Both atopic diseases and T1DM have a multifactorial etiology due to complex gene-environmental interactions. Thus, one strategy to elucidate the relationship between these two disorders is to investigate the genetic protective and risk factors associated with both of them. The genes and related molecular pathways are shown in Table 1.

Genetic analysis for *TLR2* indicated T allele in the single nucleotide polymorphism (SNP) rs3804100 as a susceptibility allele for both asthma and T1DM, and C allele as protective for both diseases [64].

Other evidence of a common genetic predisposition to asthma and T1DM comes from the GABRIEL consortium asthma genome-wide association study (GWAS), which identified 9 regions with 10 SNPs associated with asthma [65]. Among these regions, *ORLMD3/GSDMB* was the only nonhuman leukocyte antigen (HLA) region shared between childhood-onset asthma and T1DM [65,66]. It was found that rs2305480 and rs3894194, the most atopy-associated SNPs in this region (*ORLMD3/GSDMB*), were also associated with T1DM [66]. Furthermore, these two SNPs are also in high-linkage disequilibrium with rs2290400, the most associated SNP with T1DM [66]. In contrast, the minor T allele of two SNPs in *HLA-DQB1* (rs9273349 and rs1063355) seems to confer protection to both asthma and T1DM [66].

Taleb et al. [15] studied the genes implicated in the development of both asthma and diabetes, such as cytotoxic T-lymphocytes antigen 4 (*CTLA-4*) and *HLA-DQB1*0201* and *DQB1*0302*. They found an association between the G allele at the 49 (A/G) nucleotide of CTLA-4 gene and an increase in asthmatic symptoms, although not statistically significant [15]. The same 49 G allele was associated with a higher risk of diabetes mellitus in a study carried out in a Lebanese population [67]. *CTLA-4* polymorphisms might represent a common genetic risk for both diseases, although further studies are needed to confirm this hypothesis. Regarding *HLA-DQB1* alleles, a cross-sectional study in the Chinese population showed a higher frequency of *HLA-DQB1*0201* in asthmatics and a higher frequency of *HLA-DQB1*0301* in healthy controls [68]. Contrarily, Taleb et al. [15] found that diabetic carriers of the *HLA-DQB1*0201* reported significantly fewer asthmatic symptoms compared to diabetic noncarriers. A trend of higher risk of asthma symptoms was observed in diabetic carriers of *HLA-DQB1*0302* [15], although not statistically significant. The authors stated that these contrasting findings related to the development of the Th1 and Th2 immune responses were under control of HLA markers. Accordingly, the susceptible gene for one disease may become a resistant gene for the other [15].

GTPase of the immunity-associated protein (GIMAP) family proteins are modulators and regulators of the immune cell homeostasis [69]. They are highly expressed during Th1 differentiation and less during Th2 differentiation [70]. Heinonen et al. [70] carried out two Finnish population-based association studies, one for diabetes and the other for asthma and allergic sensitization, assessing the role of *GIMAP4* and *GIMAP5* polymorphisms [70]. Interestingly, the authors found that a particular *GIMAP5* SNP (rs6965571) was associated with increased risk for both asthma and allergic sensitization but was inversely associated with T1DM [70]. This association was significant only in participants from the Southwest Finland, who had a distinct genotypic structure compared to the Northeast ones. One polymorphism (rs13222905) in *GIMAP4* was only associated to both asthma and allergic sensitization [70].

The genetic findings on the association of asthma and atopy with T1DM are currently inconclusive. However, in the literature, there are promising data on the role of protective or predisposing polymorphisms involved in the development of asthma and T1DM.

A common complex polygenetic basis might exist, encouraging future association studies to better characterize the potential link between them.

The potential factors which might play a crucial role in asthma and T1DM are reported in Figure 1.

## 3. Studies Reporting no Association between Asthma and Type 1 Diabetes

In contrast to previous theories, other studies have shown no differences in asthma prevalence between T1DM subjects and the general population [26,71,72,73]. A survey of individuals with pediatric-onset T1DM in a large German/Austrian population [26] showed that the frequency of asthma in individuals with T1DM (3,4%) was similar to the prevalence of asthma in the general population (4.7% in the German population from the German KIGGS study [74] and 5.1–7.3% in the Austrian one from the ISAAC study [75]).

Jasser-Nitsche et al. [73], in an Austrian survey of 104 T1DM children and 104 healthy controls, showed no differences in allergen sensitization rate and in the number of patients reporting allergy symptoms [73].

Further studies have documented a similar frequency of allergic sensitization [71,72] and an analogue proportion of individuals reporting “lifetime asthma”, although they highlighted a lower frequency of “current asthma” and a reduced severity of asthma symptoms in the T1DM group [72]. The effect of insulin on M2 muscarinic receptors could explain this phenomenon [76]. In asthma animal models, T1DM was shown to increase the inhibitory neuronal muscarinic receptor function with an effect of reduction of airway hyperactivity [77]. On the contrary, in obese hyperinsulinemic children with asthma, the bronchial hyperresponsivity could be due to the increased acetylcholine production in bronchial smooth muscle cells by the insuline-M2 receptors interaction [78,79].

The lack of a link between these two diseases could be explained by a complex genetic-environmental interaction highlighted from studies carried out at a “country level”. Indeed, in some populations, genetic and environmental factors could counterbalance each other, leading to no differences in asthma prevalence between T1DM children and the general population.

## 4. Inflammatory Pattern of the Patients with Asthma and Type 1 Diabetes

Recent studies have investigated the cytokine profile of four different groups: Healthy children, children with only allergic diseases, children with only T1DM, and children with both diseases. In these reports, the authors substantially stated the incompleteness of the Th1/Th2 paradigm [22,80]. Klamt et al. [22] found that tumor necrosis factor alpha (TNF-α) levels were higher in children with T1DM and that IL-2 and IL-6 levels were higher in atopic children regardless of the presence of T1DM. The increase of serum levels of IL-2 in atopic children is in contrast with the Th1/Th2 paradigm [22]. On the other hand, increased levels of TNF-α and IL-6 in diabetics and allergic patients, respectively, are in agreement with the classic Th1 and Th2 pattern [22].

Rachmiel et al. [80] and Kainonen et al. [81] showed that stimulated peripheral blood mononuclear cells (PBMC) expressed a unique cytokine secretory pattern, with a combination of both Th1 and Th2 activities in patients with both asthma and T1DM. Higher serum levels of both IL-12 and IL-18 were found in asthmatics and diabetics compared to controls [80]. After stimulation of PBMCs with lipopolysaccharide (LPS) in vitro, IL-12 levels were lower in patients with both diseases compared to those with only one [80]. According to the authors, the PBMCs were “exhausted” and could not further increase the production of IL-12 in these patients [80]. No differences were shown in IL-18 levels after stimulation in vitro [80]. The IL-18/IL-12 serum ratio in vivo was also significantly higher in patients with both diseases compared to those with only asthma [80].

IL-18 is a proinflammatory key cytokine produced by dendritic cells (DC), T and B cells, and macrophages, and has pleiotropic functions interacting with various cell types [82]. Elevated IL-18 levels have been associated with asthma exacerbations and IgE levels in asthma patients [82], shown to be increased in T1DM patients, and associated with glycated hemoglobin (HbA1c) levels [38].

IL-12 is also produced by DC, B cells, and macrophages, and the main cellular targets are T cells and NK cells [83]. In T1DM pathogenesis, IL-12 and IL-18 seem to enhance cytotoxic T lymphocytes and NK cell cytotoxic activity and disrupt immunoregulation by Tregs [84].

Kainonen et al. [81] documented that the spontaneous production of INF-γ, TNF-α, and IL-10 by mononuclear cells was elevated in children with both asthma and diabetes compared to controls and patients with only one disease. In addition, the stimulation in vitro increased IL-10 secretion in solely diabetic, solely asthmatic, and healthy children but not in children with both diseases [81]. The persistency of high levels of inflammatory cytokines, despite the hypersecretion of the anti-inflammatory IL-10 [85], could indicate a deficit in the regulatory mechanisms of the inflammatory response [81]. Furthermore, Treg cells in children with both asthma and diabetes might be “exhausted” as suggested by the failure of the stimulation test in vitro in this group of patients [81]. This would mean that Treg cells in these patients are overstimulated and cannot further increase the cytokine production [81].

Infections could activate T cells against allergens or self-antigens due to the highly cross-reactivity of T cell receptors or to bystander activation. To prevent these situations, IL-10 is produced to suppress unwanted responses [86].

More recently, IL-10 has been shown to act directly on Th2 cells in a murine model of house dust mite allergy [87].

It follows that a defect of this suppressive mechanism might facilitate the development of atopic and autoimmune diseases, as shown in Figure 2.

Tregs and other IL-10-producing cells might act as the gatekeepers of the immune system, and early repeated infections would contribute to this role. Without such a stimulation, altered immune responses could cause autoimmune or allergic diseases in genetically predisposed individuals [86].

## 5. Influence of Asthma on T1DM Control and Vice Versa

A recent population-based cohort study [88], comparing a group of asthmatic diabetics with a group of age and gender-matched asthmatic nondiabetics, demonstrated that T1DM was associated with higher asthma medication use in the first year after the onset of diabetes [88]. The use of asthma medication decreased in both cohorts after the first year [88]. However, the incidence rate of asthma exacerbations was not statistically different in the two groups [88]. These data could be partly related to the frequent contact with a physician in the diabetic cohort (detection bias) [88]. However, it is also possible that the two diseases could influence each other’s control rate [88]. Black et al. [25], in American children with T1DM, reported an association between asthma and poor glycemic control, especially if asthma was untreated. The authors suggested an anti-inflammatory effect of the anti-asthma treatment, which underlined the complex relationship between pulmonary function, body mass index, and glycemic control among children with diabetes [25].

Hörtenhuber et al. [26] investigated the asthma effect on glycemic control in 52,926 patients with T1DM under 20 years of age obtained from the German and Austrian registries. Patients with asthma who received higher insulin doses experienced more severe hypoglycemia, although no differences in HbA1c were reported between subjects with and without asthma overall [26]. The authors explained the use of higher insulin doses by stress, asthma-associated inflammation, use of steroids, and less physical activity [26].

## 6. Temporal Relationship between Asthma and Type 1 Diabetes: Asthma to Type 1 Diabetes or Type 1 Diabetes to Asthma?

A recent birth cohort study of children born in Sweden examined the possibility of a shared risk of asthma and T1DM by studying their pattern of familial coaggregation [89]. This study demonstrated the co-occurrence of the two diseases with an apparent unidirectional sequential appearance. Only children who first developed asthma had a subsequent increased risk of developing T1DM (HR, 1.16; 95%CI, 1.06–1.27). In contrast, T1DM was not associated with subsequent risk of asthma (HR, 0.92; 95%CI, 0.75–1.12). Siblings of individuals with asthma or T1DM were at an increased risk of subsequent T1DM and asthma, respectively [89]. Metsälä et al. [90] showed a higher incidence of T1DM in Finnish children with asthma (0.6%) than healthy children (0.4%), whereas a lower asthma incidence was observed in children with T1DM (1.7%) than in healthy children (2.7%) [90]. The coaggregation family studies in literature stress the importance of temporal sequential appearance of autoimmune and atopic diseases [89,90]. We speculate that these findings might not be applicable to a general population because the studies were carried out only in specific ethnic groups. Further longitudinal observational studies in different ethnic and geographical contexts are needed to verify the temporal sequence.

Krischer et al. [91] demonstrated that, in children with a first-degree relative affected by T1DM, the occurrence of islet cell antibodies (ICA) as the first-appearing diabetes-related autoantibodies reduced the subsequent risk of asthma, rhinitis, and eczema. Autoantibodies to insulin (IAA) or glutamic acid decarboxylase autoantibodies (GADA) reduced the subsequent risk of eczema. When multiple autoantibodies appeared first, the reduced risk of asthma and allergic rhinitis remained significant.

On the other hand, in 7208 Swedish children, Wahlberg et al. [92] observed an association between wheezing during the first year of life and a subsequent appearance of GADA or double positivity for GADA and tyrosine phosphatase-related insulinoma-associated 2 molecule (IA-2A) at the age of 2.5 years.

## 7. Conclusions

The immunopathology of T1DM and atopic illnesses is very complex and involves different gene-environment interactions and pathways and a possible involvement of gut microbiome and dietary factors, influencing the immune response and the development of immune tolerance. Epidemiological data are inconclusive, showing controversial results about the prevalence of asthma in T1DM patients. This fact could be explained by methodological problems related to asthma diagnosis in children. Specifically, asthma diagnosis was based mainly on surveys, which were different from study to study and based on parent and self-reported questionnaires on symptoms and drug use leading to recall bias. Moreover, it is known that the diagnosis of asthma is more complicated at early ages, making it difficult to differentiate between asthma and wheezing.

Recent data on the cytokine profile of patients with both T1DM and allergic symptoms have indicated that Th1 and Th2 profiles can coexist and are not strictly mutually exclusive. These observations promote research for another immunological mechanism explaining the association between atopic and autoimmune diseases in some patients. A promising key element might be represented by the immune regulator system characterized by Treg cells, whose cytokine production has complex interaction with both Th1 and Th2 pathways. Defects in immune system response are driven by environmental factors according to the “Hygiene Hypothesis”, but also by peculiar genetic backgrounds. A better characterization of the genetic polymorphisms and the environmental factors could explain this chaotic and multifactorial potential association leading to new diagnostic and therapeutic approaches. Furthermore, more recent coaggregation studies have highlighted the importance of the temporal order of appearance of these two diseases. Asthmatic children seem to have a higher risk of developing diabetes, whereas diabetic children seem to be relatively protected from asthma development. Further epidemiological studies are needed to confirm this evidence.

Finally, following the review of available literature, we suggest investigating the glycemic metabolism in children with asthma and the lung function in children with diabetes in clinical practice.

## Figures and Tables

**Figure 1 ijms-22-05757-f001:**
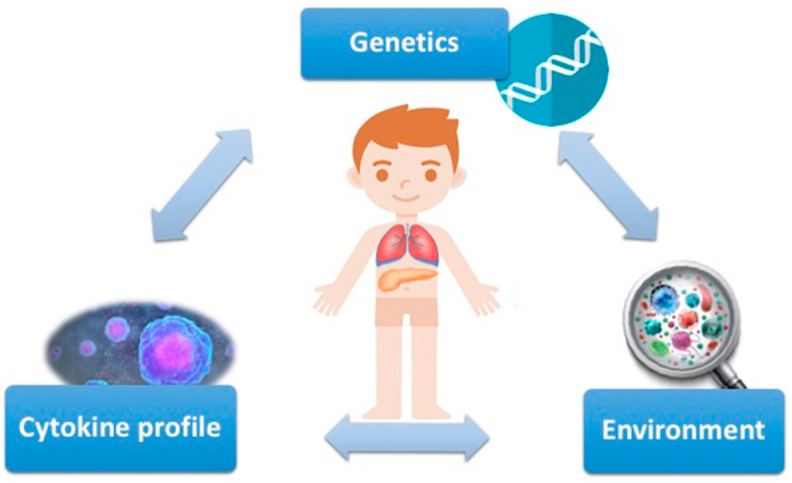
Factors potentially involved in asthma and T1DM development. Children affected by both asthma and T1DM show a unique cytokines profile. This characteristic is probably the result of a complex interaction between genetic polymorphisms and environmental factors. The multifactorial relationship should be considered as bidirectional in nature, given that every element influences the others without a clear consecutive order.

**Figure 2 ijms-22-05757-f002:**
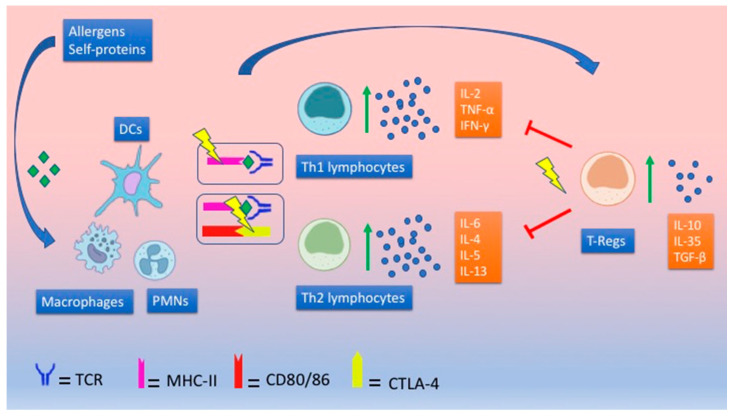
Immunological response in children with autoimmune/atopic diseases. Children developing both asthma and T1DM show high levels of both Th1 and Th2 cytokines. Anti-inflammatory cytokines, such as IL-10, are also highly expressed, even if they do not further increase after stimulation in vitro. This pattern characterized by “exhausted” Tregs might be caused by a defect in T CD4+ inhibition by IL-10, polymorphisms in HLA genes, and defects of CTLA-4 in association with environmental triggers. Yellow lightening symbols in the picture mark the possible defective sites of immunoregulation.

**Table 1 ijms-22-05757-t001:** Genes and relative molecular pathways involved in asthma and T1DM.

Gene	Protein	Function	T1DM-Associated	Asthma-Associated
*FLG*	Profilaggrin	Precursor of filaggrin, necessary for epidermal structure	No	Yes
*SEPS1*	Selenoprotein S	Translocation of misfolded proteins from the endoplasmic reticulum to the cytosol	No	Possible
*IL18*	Interleukin 18	Proinflammatory cytokine of the IL-1 family	No	Possible
*IL12RB1*	IL-12 receptor β1	Subunit of the IL-12 receptor	No	No
*IL12RB2*	IL-12 receptor β2	Subunit of the IL-12 receptor	No	No
*TLR2*	Toll Like Receptor 2	Recognition of PAMPs and activation of the innate immune response	Yes	Yes
*TLR4*	Toll Like Receptor 4	Recognition of PAMPs and activation of the innate immune response	No	No
*CD14*	Cluster of Differentiation 14	Recognition of LPS and PAMPs by macrophages	No	No
*GSDMB*	Gasdermin B	Regulation of apoptosis	Yes	Yes
*HLA-DRB1*	Human Leukocyte Antigen DR β1	Subunit of HLA class II receptor	Yes	Yes
*HLA-DQB1*	Human Leukocyte Antigen DQ β1	Subunit of HLA class II receptor	Yes	Yes
*CTLA4*	Cytotoxic T-lymphocyte-associated protein 4	Downregulation of immune responses, constitutively expressed by Tregs	Possible	Possible
*GIMAP*	GTPase of the immunity-associated protein	Regulators of immune cells homeostasis	Yes	Yes

PAMPs: Pathogen-associated molecular patterns; LPS: Lipopolysaccharide; HLA: Human leukocyte antigen.

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
