# Peer review of "Coaggregation of Asthma and Type 1 Diabetes in Children: A Narrative Review"

_ijms, 2021, doi:10.3390/ijms22115757_

Round 1

Reviewer 1 Report

In this manuscript, the authors reviewed the scientific evidence present in literature on the association between asthma and T1DM in childhood as well as discuss the possible mechanisms underlying this relationship, such as genetic predisposition, cytokine patterns and environmental influences. Overall, this review is still incremental work without much novelty and update of the field. Concerns listed below should be carefully addressed.

  1. Many references are still outdated, the authors should consider more recent works.
  2. I think the authors need to stress that this manuscript only suggests that T1DM and asthma are not mutually exclusive. Correlations/associations are not confirmed. However, in section 2 and later, the authors seem to focus more on a “forced” association between T1DM and asthma which is too speculative (without much evidence). The whole manuscript needs to be restructured.

Reviewer 2 Report

After a comprehensive review of this manuscript, unfortunately, I do not recommend it be accepted for publication. This manuscript requires extensive editing and restructuring of the English language and style. Throughout, some parts do not make sense or the incorrect word has been used to describe what is being discussed. Also throughout, there are statements made that are not supported by references and in many instances, outdated references have been used. I also have concerns that the authors have not correctly interpreted the literature and in parts provide very basic descriptions of key concepts that are not of the standard for publication. Several reviews on the same subject have been published in the last few years. The present manuscript doesn't seem to add any novel insight.

Round 2

Reviewer 1 Report

After reading the authors' reply, I feel that the authors still do not have a clear purpose in writing this review. I understand that the authors want to consolidate previous research results in this area. However, it is clear that asthma and type 1 diabetes have little or no association and hence, not a suitable topic to be written as a review. Failure to find sufficient and major original research publications in the past 5 years also indicates that this area is no longer an interesting topic for further investigation. Therefore, I cannot recommend publication of this manuscript.

Reviewer 2 Report

The authors satisfactorily addressed all criticisms raised during the peer-review process and improved manuscript quality. The revised manuscript is now worthy of publication.

This manuscript is a resubmission of an earlier submission. The following is a list of the peer review reports and author responses from that submission.

Round 1

Reviewer 2 Report

In this paper , the authors reviewed the relationship of asthma and Type 1 diabetes in children, and discussed the current theories as well as the possible factors which may affect the incidence of allergy disease in T1DM. The review is comprehensive, only with a few questions should be addressed.

  1. Each topic discussed the current progresses of asthma and T1DM, but lack of summary discusses.
  2. The paragraph number of 2.1.1 should be 2.2.1
  3. There should be a table or graph to display all the genes and related molecular pathway that related with T1DM and asthma.
  4. “Gut microbiome” paragraph: this paragraph only discussed a little about gut microbiome and T1DM, but not asthma andT1DM.
  5. There are many controversial results about the incidence of asthma in T1DM patients, so it should be better to discuss more about the reasons that caused those discrepancy.

Reviewer 3 Report

In this manuscript, the authors reviewed the scientific evidence present in literature on the association between asthma and T1DM in childhood as well as discuss the possible mechanisms underlying this relationship, such as genetic predisposition, cytokine patterns and environmental influences. Overall, this review is incremental work without much novelty and update of the field. Concerns listed below should be carefully addressed.

  1. The title of this manuscript can be edited to be “Coaggregation of Asthma and Type 1 Diabetes in Children: A Narrative Review”.
  2. There are many recent publications on the global trends of diabetes and asthma. Yet, the authors only cited publications more than 10 years ago in the introduction part regarding global trends.
  3. Only about 20% of the cited work are less than 10 years old (even less work within 5 years old). Many cited publications are  also from the 1990s. The authors should read and cite more recent work in this field since the purpose of a review is to consolidate and discuss recent updates in the field.
  4. Language use in this manuscript is inappropriate and should be improved. For example, in line 403-404, “following our revision of literature we would suggest the importance” should be “following the review of available literatures we would like to suggest the importance”. Proofreading of the manuscript by a native English speaker is highly recommended.

A very major concern for this review manuscript is that it does not adequately update the reader regarding this field as over 80% of the publications cited are severely outdated. Therefore, I cannot accept this manuscript in the current state.